# A Permanent Magnet Synchronous Spherical Motor for High-Mobility Servo-Actuation

**Jay A. Shah, Samuel R. Miller, Shaphan R. Jernigan and Gregory D. Buckner ***

Mechanical and Aerospace Engineering, North Carolina State University, Raleigh, NC 27695, USA; jshah4@ncsu.edu (J.A.S.); srmille7@ncsu.edu (S.R.M.); srjernig@ncsu.edu (S.R.J.)
* Correspondence: gbuckner@ncsu.edu

**Abstract:** The development of direct-drive spherical motors offers a potential solution to the limitations of conventional multiple degree-of-freedom (DOF) actuators, which typically utilize single-DOF joints (rotational and/or prismatic), arranged in series or parallel and powered by multiple single-DOF actuators. These configurations can be accompanied by kinematic singularities, backlash, limited power density and efficiency, and computationally expensive inverse kinematics. This paper details the design, fabrication and experimental testing of permanent magnet synchronous spherical motors (PMSSM) for multi-DOF servo-actuation. Its stator-pole arrangement is based on a Goldberg polyhedron, with each pole comprised of hexagonal or pentagonal inner and outer plates. The stator geometry and winding configurations are optimized using electromagnetic finite element analysis. A custom-made controller board includes a microcontroller, servo drivers, a wireless serial interface, and a USB PC interface. Angular orientation is sensed using an inertial measurement unit in wireless communication with the microcontroller. A PID controller is implemented and demonstrated for time-varying reference trajectories.

**Keywords:** spherical motor; permanent magnet synchronous motor; PID control





## 1. Introduction

As servomechanism technologies and applications continue to advance, there is a growing need for innovative actuators with increased kinematic mobility and power density. Servomechanisms with multiple degrees-of-freedom (DOF) are typically realized using single-DOF joints (rotational and/or prismatic), arranged in series or parallel and powered by multiple single-DOF actuators. These configurations can be accompanied by kinematic singularities, backlash, limited power density and efficiency, and computationally expensive inverse kinematics. The development of direct-drive spherical motors offers the potential means to overcome these limitations by replacing multiple single-DOF actuators with a single multi-DOF actuator. Potential applications include satellite attitude control [1], encoder, camera and scanner gimbals [2], rotational stages for CNC machining and additive manufacturing [3], haptic feedback devices [4], and robotic actuation [5,6].

The research and development of spherical motors spans several decades and varies widely in the literature. Common characteristics of almost all spherical motors are a stator and a rotor with three angular DOF (simultaneous rotation about three orthogonal axes). Significant design variations address methods for rotor support, actuation, and orientation sensing. Most spherical motors utilize ball-and-socket joints [1,6–9] for accommodating multi-DOF motion, although three-axis gimbals and other novel linkage configurations can be found in the literature [10]. An advantage of the gimbal architecture is that orientation sensing can be readily achieved via inverse kinematics of cascaded single DOF encoders [2]. However, the added weight of gimbal linkages and bearings reduces power density, and kinematic singularities (e.g., gimbal lock) limit the capabilities of these systems. Ball-and-socket joints are less kinematically problematic than gimbaled systems (i.e., are not prone

to gimbal lock) but require spherical bearing surfaces, which can be difficult to fabricate and implement with sufficient accuracy.

While most spherical motor prototypes are actuated electromagnetically, typically via alternating currents within stator coils, a small number rely on induction between the stator and rotor [1,6,11,12]. Vachtsevanos conceptualized and rigorously analyzed a spherical induction motor, but the complex design proved impractical to prototype. Kumagai and Hollis fabricated a spherical induction motor with a potential application in robot locomotion, and experimentally demonstrated its operation via closed-loop control. While its torque production was high by comparison to other spherical motors (4 N·m), a key deficiency was its low efficiency, likely a consequence of eddy currents within the unlaminated iron rotor.

Synchronous AC spherical motors are the most common motors reported in the literature [2,7,13–15]. Most incorporate permanent magnets in the rotor, a notable exception being the variable reluctance spherical motor studied extensively by [8], which features a solid iron rotor. The arrangements of stator coils and rotor poles (typically permanent magnets, PMs) varies widely among synchronous spherical motors, as do the materials for these components. Despite these differences, most feature cylindrical rotor and stator poles.

Another intriguing actuation method for spherical motors involved piezoelectric materials. Studied by multiple researchers since the 1990s, spherical ultrasonic motors (SUMs) employ mechanical interaction between the rotor and piezo-actuated stator components. Although working SUM prototypes have been demonstrated, their limitations include low torque production [5,16,17], frictional wear, low rotational speed, and hysteretic behavior.

While gimbal-based spherical motors employ traditional single-DOF bearings, omnidirectional bearings are needed for ball-and-socket configurations. Perhaps the simplest omnidirectional bearing is the ball transfer, also referred to as a Hudson bearing. A variant of the ball transfer is the spring plunger. These bearings consist of a partially enclosed rotating sphere, which supports the rotor via direct contact and provides low-friction relative motion. Ball transfers are commonly used due to their simplicity and commercial off-the-shelf availability in a wide variety of sizes and form factors [6,9,14]. Limitations of this bearing include relatively low load capacity and relatively high friction compared to other methods [18]. Fluid bearings provide another omnidirectional support option but are inherently more complex than ball transfers. Week, et al. [7] proposed a spherical motor with a hydrostatic bearing consisting of multiple oil-filled pockets. A hydraulic flow controller at each pocket was proposed to provide appropriate pressure at each pocket, according to its loading requirements. Although plans to fabricate a prototype were mentioned, no follow-up publication of outcomes could be found in the literature. Similarly, Ezenekwe and Lee [19] analytically modeled a spherical motor/actuator with an air bearing system, but did not report results from a working prototype. Tan and Huang [20] designed and fabricated a spherical air bearing and demonstrated 2-DOF control of the device. The air bearing utilized a thin film of pressurized gas to suspend the rotor within the stator and eliminate friction between the two. Among other components, it included an air compressor, air tank, a series of particle filters, a pressure regulator, vacuum generator, and air supply lines.

Closed-loop control of spherical motors requires a method for sensing relative orientations between the rotor and stator. As mentioned previously, gimbaled systems can readily accommodate rotary encoders on each axis to facilitate orientation measurements [2]. Some ball-and-socket systems incorporate rotating slides attached to rotary encoders. These systems accommodate only limited angular rotations [7,16,21] and add mass to the rotor, reducing its power density. A limited number of prototypes utilize multiple Hall-effect sensors on the stator [5,8], but have limited practicality for AC excitation that would require large quantities of rotor PMs and high magnitudes of rotation. Kumagai and Hollis's induction motor incorporated four optical mouse sensors at the interface between the stator and rotor to measure rotor surface displacements. Other methods include a dual-axis

position sensing diode [20] and an optical sensor specifically developed for spherical motor orientation sensing [22].

## 2. Materials and Methods

### 2.1. Stator Design

While the stators of conventional (1D) electric motors utilize large numbers of closely spaced axial slots and windings, each having identical geometry, such design approaches cannot be applied to spherical motors. Unlike conventional electric motors, which typically use radial patterns for poles and windings, the PMSSM pole-winding configuration is non-trivial. To accommodate near-spherical geometry, the stator poles of the PMSSM were arranged to form icosahedrons, including a subclass of Goldberg polyhedrons comprised of pentagons and hexagons. Goldberg polyhedrons are represented as GP(m,n), where m represents the number of steps in the given direction, and n represents the steps following a 60° turn from one pentagon to the next. For the PMSSM stator geometry, a GP(3,0) consisting of 92 discrete faces (12 pentagons, 20 regular hexagons and 60 irregular hexagons) was chosen (Figure 1). This geometry is desirable because it minimizes differences in surface area between the smallest and largest polygon pole faces.

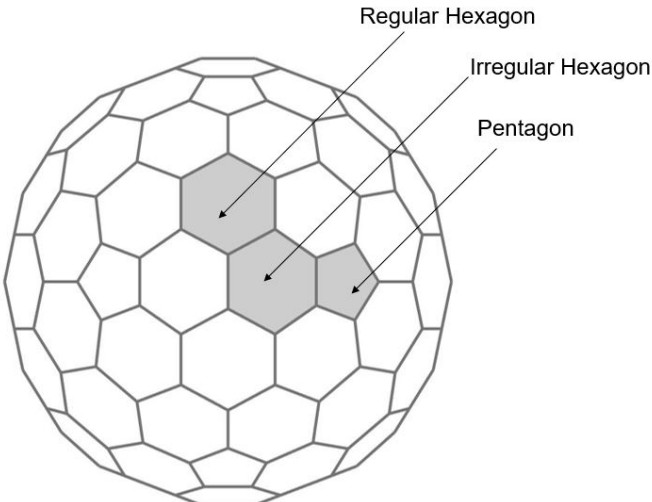

**Figure 1.** Goldberg polyhedron (3,0) used as the basis for PMSSM stator design. Highlighted are the fundamental geometries: pentagon (12 total), regular hexagon (20 total) and irregular hexagon (60 total).

Because the PMSSM stator must permit a specified range of motion for the rotor and its coupled load (in this case +/−40°), the stator geometry is a hemisphere comprised of only 46 (of 92 possible) poles. Each stator pole consists of a top plate, a stem and a bottom plate. The top plates of adjacent stator poles are separated by small air annular gaps (2 mm), while the bottom plates are in direct contact to provide low reluctance return paths for magnetic flux. Figure 2a shows an assembly of three adjacent stator poles (a regular hexagon, an irregular hexagon, and a pentagon), while Figure 2b shows the key dimensions of a pentagonal stator pole. The stator poles were manufactured using 1018 cold rolled steel, due to its machinability, high magnetic permeability, and low cost.

The stator pole dimensions were iteratively optimized to provide a peak stem flux density, *B*, in excess of 1.0 T, while simultaneously minimizing pole materials (steel and magnet wire). The specified PMSSM rotor diameter ($D_r$ = 152.4 mm) and stator/rotor air gap (g = 2.0 mm) enabled direct calculation of each major diameter $p_i$ [23]. The stem diameter *d*, stem length *l* and plate thickness *t* were optimized to provide adequate volumes for the copper windings, while simultaneously restricting the outer diameter of the stator ($D_s$ = 213.5 mm). To prevent flux saturation in the top and bottom plates, thickness t must

be large enough to provide adequate cross-sectional areas for the flux produced in one stem to return through the neighboring stems.

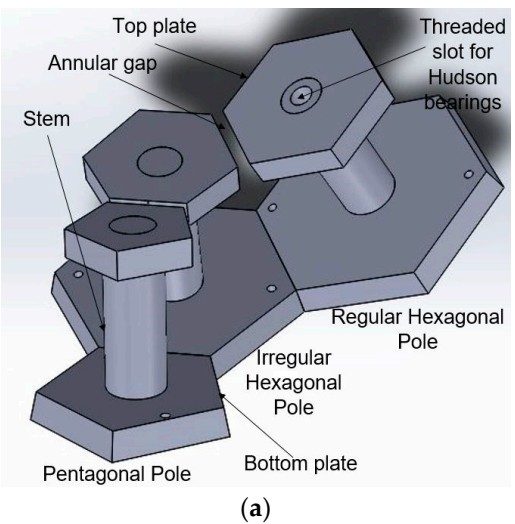

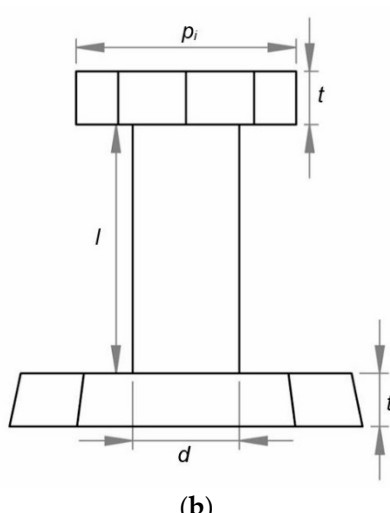

(**a**)                                                              (**b**)

**Figure 2.** Stator pole geometry: (**a**) solid model of three adjacent stator poles showing annular gaps between top plates and direct contact between bottom plates; (**b**) pentagonal pole dimensions: top and bottom plate thickness *t*, stem length *l*, stem diameter *d* and major diameter of polygon face.

$$\pi d t \geq \frac{\pi d^2}{4} \tag{1}$$

Based on orthocyclic windings (which have a packing factor of 90.7% [24]), the number of turns $N$ in a fully wound coil of magnet wire (diameter $d_w$, packing factor $p$, and rectangular cross-sectional area of width $b$ and height $h$) can be computed as the following:

$$N = \frac{4bhp}{\pi d_w{}^2} \tag{2}$$

To optimize these stator pole dimensions, three-dimensional magnetostatic simulations were conducted using finite element analysis (FEA) software (ANSYS Electronics Desktop, Canonsburg, PA, USA). Each simulation utilized more than 300,000 tetrahedral elements, with curvilinear meshing automatically generated. Parametric sweeps quantified the nonlinear relationships between magnetic flux density and stator pole dimensions, most notably stem diameter $d$ and length $l$. Figure 3 shows the FEA results for the adjacent stator poles simultaneously excited with the maximum 300 A-t; case 3b ($d = 12$ mm) effectively addresses the tradeoff between stator stem mass and magnetic saturation. While the stator pole excitations used in these simulations do not necessarily correspond to those used for rotor torque production, they provided useful insights into dimensional optimization.

Similar parametric sweeps were used to optimize the remaining stator pole dimensions, as summarized in Table 1. The regular hexagons are the largest in size, followed by the irregular hexagons and the pentagons.

**Table 1.** Optimized stator pole dimensions (mm).

| Dimension | Irregular Hexagon | Regular Hexagon | Pentagon |
|:---:|:---:|:---:|:---:|
| *p* | 33.32 | 37.77 | 26.21 |
| *d* | 12.00 | 12.00 | 12.00 |
| *l* | 28.50 | 28.33 | 29.71 |
| *t* | 6.35 | 6.35 | 6.35 |

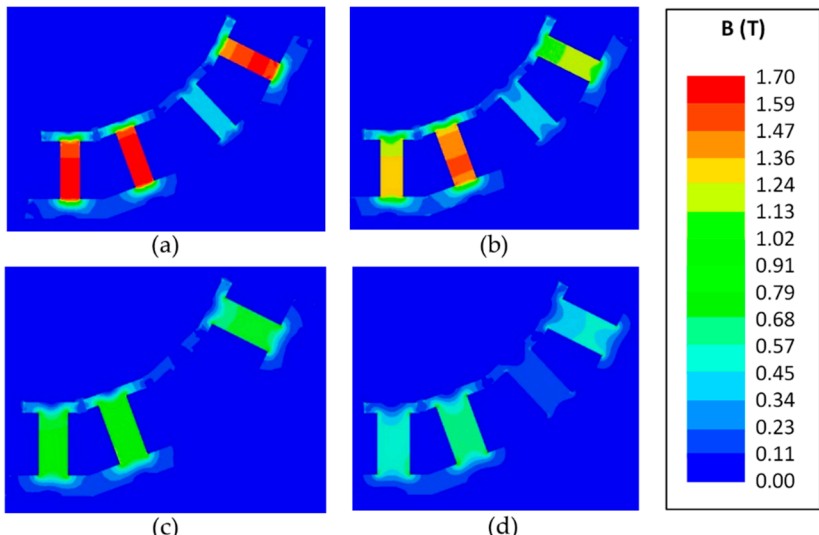

**Figure 3.** FEA simulation results showing magnetic flux density in adjacent stator stems, simultaneously excited (300 A-t), as functions of stem diameter: (**a**) $d$ = 10 mm, (**b**) $d$ = 12 mm, (**c**) $d$ = 14 mm, (**d**) $d$ = 16 mm.

### 2.2. Stator Fabrication

The top and bottom stator pole plates and stems were individually fabricated with waterjet cutting and machining process and assembled using 3M Scotch Weld MC100 adhesive (Figure 4a). The stems of regular hexagon poles (10 total) were drilled and tapped to accommodate the M6 threads of Hudson bearings (MP-6, NBK, Seki, Gifu, Japan) to ensure low-friction support of the rotor (Figure 4b). These Hudson bearings have a static and dynamic load bearing capacity of 29 N and 9.8 N, respectively, which will be shown in the Rotor Methods section.

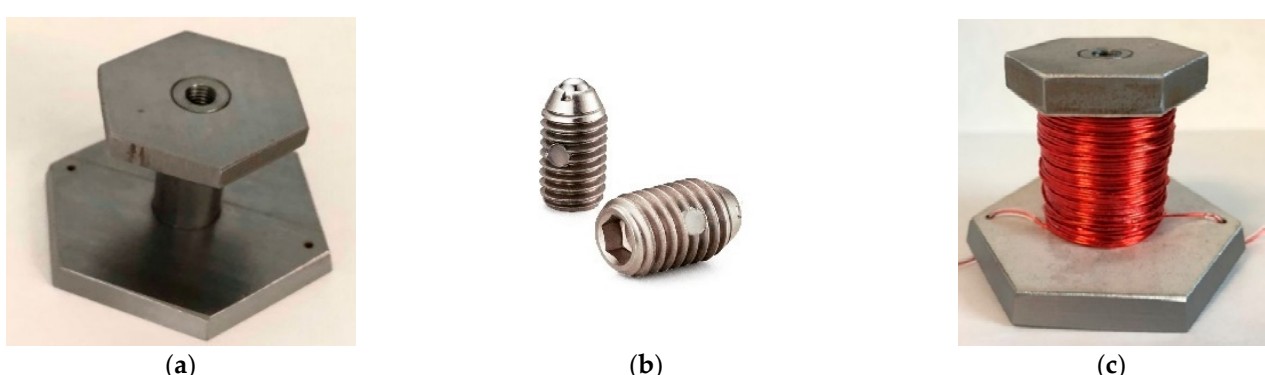

**Figure 4.** Stator poles and bearings: (**a**) assembled regular hexagonal pole unit showing tapped M6 threading for Hudson bearings; (**b**) M6 Hudson bearings assembled into the hexagonal poles; (**c**) wound stator pole ($N$ = 522 turns of 24 AWG magnet wire).

Prior to winding, the stems and plates of each stator pole were lined with 0.25 mm insulation (CG100 Fish Paper, 145PTags.com, Maitland, FL, USA) to prevent abrasion and shorting of the magnet wire. Each stator pole was wound with 522 turns of 24 AWG magnet wire (Remington Industries, Johnsburg, IL, USA), with a tabletop CNC milling machine (Sherline 2000, Vista, CA, USA) and a custom 3D-printed fixture. Two small holes (2 mm diameter) drilled into the bottom plates of each stator pole allowed the winding terminations to pass through the bottom plate of the stator pole to electrical terminations outside the stator housing. A photograph of a completed regular hexagonal pole is shown in Figure 4c.

A hemispherical stator housing (Figure 5) was designed to maintain support and proper spacing for the 46 stator poles and organize the phase conductors. This housing (Figure 5a) was 3D printed using fused deposition modeling (F370, Stratasys, Eden Prairie, MN, USA) with ABS material. The stator poles were secured into the housing using a top ring assembly and M6 bolts (Figure 5b). Through-holes in the housing (92 total) enabled external stator phase connections (Figure 5c).

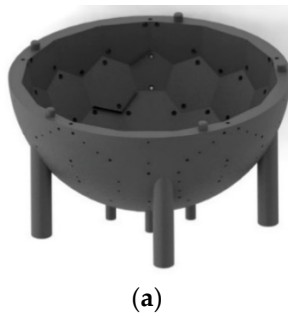

(**a**)

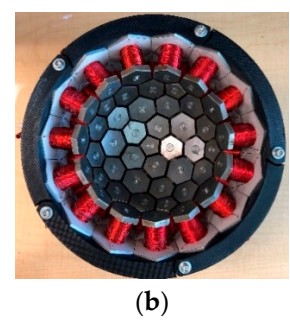

(**b**)

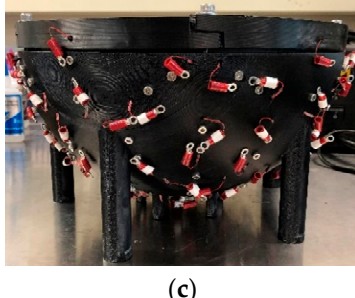

(**c**)

**Figure 5.** Stator housing: (**a**) solid model showing basic geometry; (**b**) photograph showing secured stator poles; (**c**) photograph showing external phase connections.

### 2.3. Rotor Design

The design process for the PMSSM rotor addressed unique challenges associated with its geometry. This included optimizing the number, location, size and polarity of internally-mounted permanent magnets (PMs), while providing a smooth, thin outer shell to contact the Hudson bearings for rotor support. These design parameters were optimized to provide smooth and adequate torque without excessive mass and without exceeding the bearings' load capacity (29.0 N static, 9.8 N dynamic). The rotor's outer radius was specified during the stator design process to provide a nominal air gap of 2 mm, $D_{or}$ = 154.4 mm.

Grade N42 neodymium-iron-boron magnets (NdFeB) were selected for the rotor, based on their extremely high energy density (42 MGOe) and widespread availability in common shapes and sizes. To optimize their size, number, location and polar orientations, magnetostatic simulations were performed using FEA software (ANSYS Electronics Desktop, Canonsburg, PA, USA). These simulations helped balance the tradeoffs between motor torque and normal bearing loads. Figure 6 illustrates the relationship between magnet size and normal bearing load for maximum stator pole excitation (300 A-t).

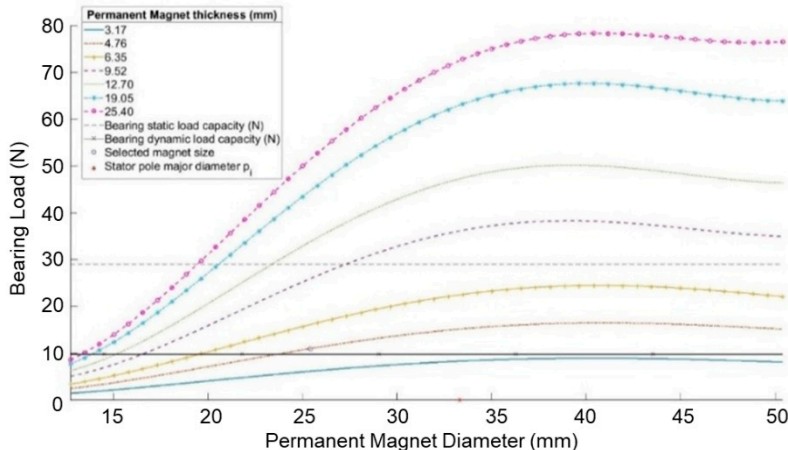

**Figure 6.** Normal bearing force as functions of diameter and thickness for cylindrical N42 NdFeB PM centered over a stator pole with maximum excitation of 300 Amp-turns. Static and dynamic load capacities for the Hudson bearing (MP-6, NBK, Seki, Gifu, Japan) are indicated.

These simulation results reveal that bearing loads increase steadily with PM diameter and thickness up to and beyond the stator pole major diameter $p_i$ (33.32 mm). Based on these results, a PM diameter of 25.4 mm and thickness of 4.76 mm were chosen to maximize motor torque, without exceeding the bearing's static load capacity at peak stator pole excitation.

After specifying PM size, algorithms were developed to optimize the number and uniform spacing of rotor PMs; details on these algorithms can be found in Shah's work [25]. One algorithm uniformly distributed a specified number (*n*) of PMs on the rotor's interior surface, while the other calculated bearing loads associated with PM-stator pole interaction forces as functions of rotor orientation. Various methods for uniformly distributing points on a spherical surface have been discussed in the literature; these include replicating Goldberg polyhedron geometries [26], minimizing the distance between points based on a Coulomb's law analogy [27], and employing Voronoi diagrams [28] and spiral schemes [29]. Goldberg polyhedrons impose geometric restrictions on the values of *n*, while Voronoi diagrams and Coulomb's law analogies are computationally expensive. Spiral schemes failed to generate uniform distributions for *n* < 32. These limitations prompted the utilization of the nearest neighbor method (NNM) [30] for *n* > 4 PMs.

Figure 7a shows an initial random distribution of 34 PMs on the surface of the rotor before location optimization. Figure 7b shows the optimized PM distribution after 2100 algorithm iterations, upon which the associated cost function had declined from an initial value of 48.01 to 0.07 for the three nearest neighbors. It is worth noting that, due to rotational limitations of the rotor, PMs mounted near the top of the rotor (specifically, located within 0.38 Steradians of the z-axis, Figure 7c) do not interact with stator poles, and hence do not contribute to torque production.

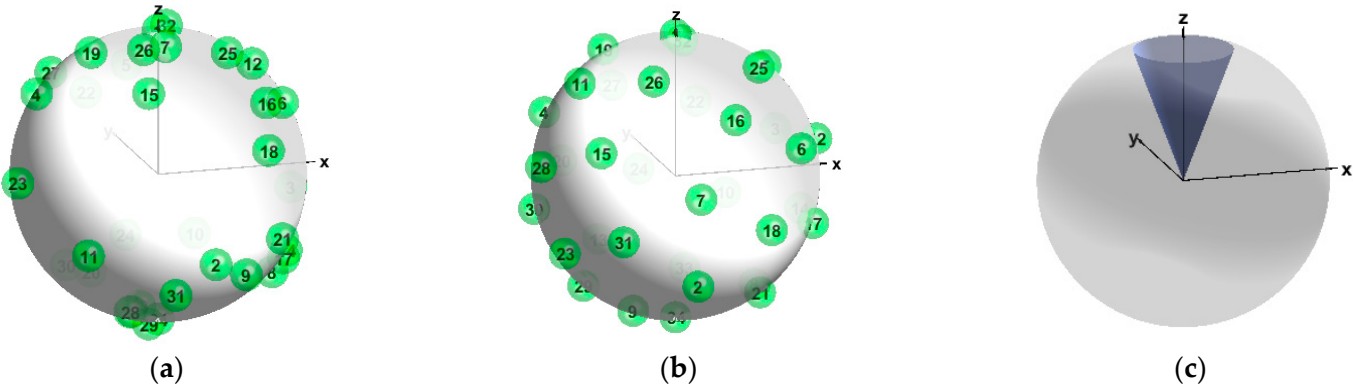

**Figure 7.** Permanent magnet placement on the rotor: (**a**) initial random distribution of 34 PMs on the surface of the rotor sphere; (**b**) distribution of 34 PMs after 2100 iterations of the NNM algorithm; (**c**) region of 0.38 sr designated for PM removal.

After uniformly distributing the *n* PMs, PM-stator pole interaction forces were calculated for a full range of rotor orientations. For normal bearing loads, the highest load scenario was a PM positioned directly over a stator pole with maximum current density (300 A-t). The interaction forces were computed along each axis, with total directional forces represented by $F_x$, $F_y$ and $F_z$. The ten Hudson bearing locations were designated $P_{kx}$, $P_{ky}$, and $P_{kz}$, where $k$ represents the Hudson bearing index. Equation (4) shows the relation between the total forces on the bearings, the bearing reaction forces ($RF_{01} \ldots RF_{10}$) and the normalized bearing locations.

$$\begin{bmatrix} P_{1x} & \ldots & P_{10x} \\ P_{1y} & \ldots & P_{10y} \\ P_{1z} & \ldots & P_{10z} \end{bmatrix} \begin{bmatrix} RF_{01} \\ \ldots \\ RF_{10} \end{bmatrix} = \begin{bmatrix} F_x \\ F_y \\ F_z \end{bmatrix} \tag{3}$$

The bearings can only provide compressive loads; this condition is represented by Equation (4), which is as follows:

$$RF_{01}, \ldots, RF_{10} \geq 0 \tag{4}$$

The optimal number of PMs (*n*) was determined by minimizing a cost function, *f*, of the bearing forces (Equation (5)) subject to the compressive force constraint (Equation (4)), using MATLAB's '*fmincon*' algorithm (Mathworks Inc., Natick, MA, USA). A factor of safety of 2.0 was added to the bearing capacity to account for the weight of the rotor and to prevent mechanical failure.

$$f = minimize\left( \sqrt{\frac{RF^2_{01} + \ldots + RF^2_{10}}{10}} \right) \tag{5}$$

The final rotor design task was specifying the polarity of each PM. The spherical geometry of the spherical motor creates a challenge to generating any symmetric alternating pattern of opposing polarities in a PMSSM, as compared to a conventional electric motor. A genetic algorithm (GA) [31,32] was implemented to obtain a uniform distribution of polarities. The cost function of the GA was a metric to track the polarities of each magnet location in accordance with its neighbors. Each magnet location was assigned an initial cost function value of zero. The value was incremented by −1 for a neighbor with an opposing polarity and + 1 for a magnet location with the same polarity. The final cost function of every design is the sum of the individual cost functions of all the magnet locations.

The results of the GA included multiple designs with the same cost function value. To further evaluate these designs, the different magnet locations were divided into five sectors based on the value of their z coordinates. Figure 8a shows the rotor divided into different sectors (*s*1, *s*2, *s*3, *s*4 and *s*5). Each sector was assigned a weight (0.15, 0.25, 0.3, 0.2, 0.1, respectively) based on its size and the duration of interaction with the stator poles (the magnet located in rotor sector S1 interacts continually). The individual cost function at each location was multiplied by its sector weight, and the cost function was updated. The cost function with the lowest value provided the solution for the best distribution of magnet polarities. Figure 8b shows the optimized polarity distribution for 11 magnets on the surface of a sphere.

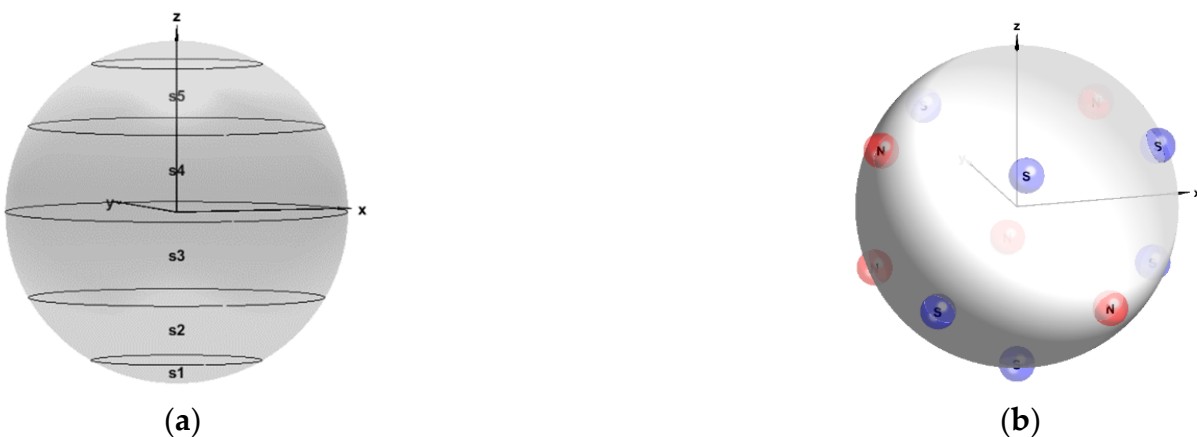

(a)            (b)

**Figure 8.** Specifying rotor PM polarities: (**a**) rotor sectors S1 to S5; (**b**) optimal distribution of magnet polarities for 11 magnet locations after removing locations in a solid angle of 0.3789 sr about the z-axis.

The optimized rotor is comprised of 11 N42 NdFeB magnets 25.4 mm in diameter and 4.76 mm thick (Applied Magnets, Plano, TX, USA). The normalized magnet locations and polarities are provided in Table 2.

**Table 2.** Normalized $x$, $y$, z coordinates and polarity of rotor PMs.

| | | | | | | | | | | | | |
|---|---|---|---|---|---|---|---|---|---|---|---|---|
| **Coordinates** | $x$ | 0 | 0.35 | 0.2 | 0.89 | −0.77 | −0.68 | 0.68 | −0.20 | −0.89 | −0.35 | 0.76 |
| | $y$ | 0 | −0.82 | 0.87 | 0.08 | 0.46 | −0.59 | 0.59 | −0.87 | −0.08 | 0.82 | −0.46 |
| | $z$ | −1.00 | −0.45 | −0.45 | −0.45 | −0.45 | −0.45 | 0.45 | 0.45 | 0.45 | 0.45 | 0.45 |
| **Polarities** | | S | N | N | S | N | S | N | S | N | S | S |

### 2.4. Rotor Fabrication

To facilitate manufacturing, and to allow access to the inertial navigation system (VN-100, Vectornav, Dallas, TX, USA), the rotor was comprised of two parts, the rotor bowl and the removable lid. The bowl housed the 11 PMs and provided a hard, smooth bearing surface, while the lid housed the electronics used for orientation sensing. The rotor bowl consisted of an inner support structure (Figure 9a), which was 3D printed using the same equipment and material as that of the stator housing, outlined above. An outer layer of hard epoxy resin (EpoxAcast 690, SmoothOn, Macungie, PA, USA) was cast around this inner structure using a two-part mold that was 3D printed using the same FDM material (Figure 9b). After removing the mold, this rotor was machined to final dimensions using a CNC lathe (Prototrack 1840 SLX, Southwestern Industries, Los Angeles, CA, USA).

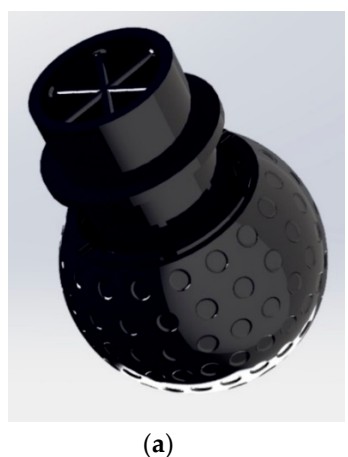

(**a**)

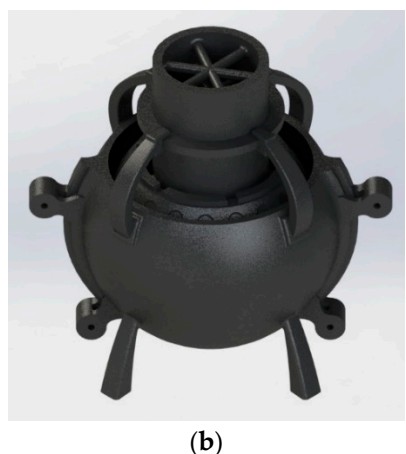

(**b**)

**Figure 9.** Fabrication of the rotor bowl: (**a**) the ABS inner structure of the rotor around which epoxy is cast; (**b**) the two-part mold assembly surrounding the inner structure.

The PMs were secured to the inner rotor bowl (Figure 9a) using an adhesive epoxy (Clear Multi-Purpose, Loctite, Düsseldorf, Germany). The rotor lid houses the IMU, the RF transciever module (Digi International, MN, USA), and associated electronics. The rotor bowl and the lid are assembled with an interference fit to form a complete sphere. The final PMSSM prototype, detailing the fully assembled rotor and stator sub-assemblies, is shown in Figure 10.

### 2.5. Control Theory

To enable orientation control, the rotor orientation and the orientation error must be defined. The 3D orientation of the PMSSM can be represented using Euler angles; the device's rotation about three orthogonal axes is represented by angles $\alpha$, $\beta$ and $\gamma$, as shown in Figure 11.

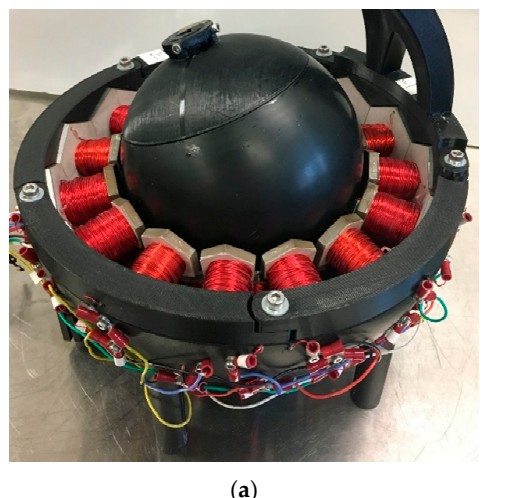 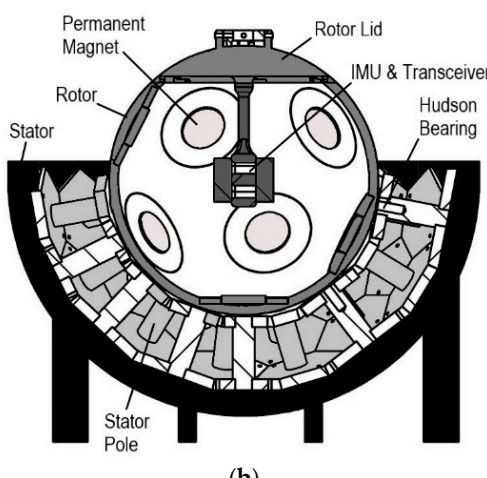

| (**a**) | (**b**) |
|---|---|

**Figure 10.** Complete PMSSM assembly: (**a**) Photograph and (**b**) Cross-sectional schematic.

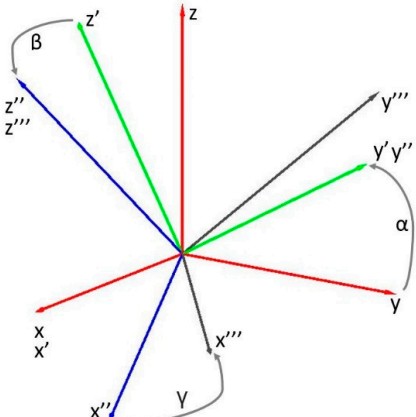

**Figure 11.** Euler angles $\alpha$, $\beta$ and $\gamma$ used for quantifying angular rotor displacement.

Quaternions were utilized, as they provided singularity-free representations (unlike Euler angle representations). Details involving the relationships between quaternions, Euler parameters, and Euler's axis-angle theorem can be found in Shah's work (2020). The orientation error vector can be expressed in inertial (fixed) coordinates as $\hat{e} = e_x\hat{\imath} + e_y\hat{\jmath} + e_z\hat{k}$; it represents the axis about which the rotor must be rotated to achieve the desired orientation. To design a proportional orientation controller for this system, the desired torque vector $\vec{\tau}_d$ should be pointed along $\hat{e}$, as this would result in rotor motion towards the desired trajectory. Such a proportional controller could be implemented in the body (moving) coordinate system, shown by the following equation:

$$\vec{\tau}_d = K_P\left(e_1\hat{\imath}_{\overline{B}} + e_2\hat{\jmath}_{\overline{B}} + e_3\hat{k}_{\overline{B}}\right) \tag{6}$$

where $K_P$ is the proportional controller gain. If $\vec{\tau}_d = d_1\hat{\imath}_{\overline{B}} + d_2\hat{\jmath}_{\overline{B}} + d_3\hat{k}_{\overline{B}}$, this approach can be extended to proportional + integral + derivative (PID) control by

$$d_i = K_P e_i + K_I \int_0^t e_i(\tau)d\tau + K_D\dot{e}_i, \quad i = 1, 2, 3 \tag{7}$$

where $K_P$, $K_I$, and $K_D$ are the proportional, integral, and derivative gains, respectively. The phase currents that result in the desired torque, $\vec{\tau}_d$, can be computed using the characteristic torque relation [33]; $\vec{\tau}_m = KL\vec{u}$ where $K$ is the characteristic torque matrix, $L$ is the phase

group matrix, and $\vec{u} = \begin{bmatrix} I_1 & I_2 & \dots & I_9 \end{bmatrix}^\top$ is a vector of phase currents. This results in the following control law:

$$\vec{u} = (KL)^\dagger \vec{\tau}_d \tag{8}$$

where $(KL)^\dagger$ is the least-squares pseudoinverse, which can be calculated using the right inverse.

### 2.6. Control Hardware

The controller is implemented on a custom-made circuit board, which consists of a microcontroller unit (MCU), servo drivers, a wireless serial interface, and a USB interface. The servo drivers utilized the DRV8801 module (Texas Instruments, Dallas, TX, USA). Each module uses N-channel power MOSFETs in full H-bridge configuration, which can drive each motor phase with peak maximum currents of 2.8 A at up to 36 V. The VN-100 IMU was used to measure the motion and orientation of the rotor in real-time. This sensor's capabilities include onboard sensor fusion, calibrated measurements with accuracies exceeding 0.5° in pitch/roll, and a customizable serial interface that can directly output relevant information, such as quaternion orientation and angular velocity. To communicate with the IMU without interfering with the motion of the rotor, a wireless serial interface was established using XBee S2C transceiver modules (Digi International, Hopkins, MN, USA). An STM32F407VG (STMicroelectronics, Geneva, Switzerland) MCU was used to capture current and IMU sensor readings, perform control algorithm computations, and control each EM driver. In addition, a USB interface was used between the MCU and a desktop PC for viewing and recording data.

The 46 stator poles were connected in individual phases similar to conventional electric motors. However, the stator's Goldberg polyhedron geometry and the multi-DOF nature of the rotor implied that the stator could not be wound in conventional 3-phase groupings. The stator poles were grouped in phases based on the geometry of the Goldberg polyhedron and ensuring that no two consecutive stator poles are of the same phase. Figure 12 shows the five different geometric patterns used to wind all the stator poles into phases.

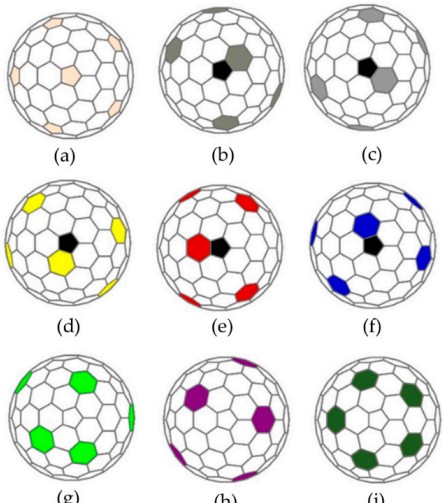

**Figure 12.** Geometric representation of stator pole phases: (**a**–**i**) represent phases 1–9, respectively. Phases 2–6 are symmetric about the edges of the central pentagon highlighted in black.

Because stator poles in the same phase can be excited with opposing magnetic polarities, these polarities should be chosen to maximize synnergistic efforts between stator poles in the same phase. For example, the five stator poles in Phase 9, as shown in Figure 12e, can each be connected with either a northward or southward polarity. If the torque vectors generated by any pair or stator poles in the same phase tend to point in the same direction, then the poles should have the same polarity. However, if the torque vectors tend to point in opposite directions, then the poles should have opposite polarities. This helps

ensure that all poles in a given phase are working in unison to produce the largest net torque possible.

This synnergistic effort can be quantitatively assessed by computing the inner product of the characteristic torque vectors ($\vec{K}_i$ and $\vec{K}_j$), summed over a subset of possible rotor orientations ($q$). This is shown by the finite sum in Equation (9) below. Here, $S_q$ represents the set of all possible rotor orientations from which individual orientations are selected. These orientations are selected so that the inner product is summed over the expected range of motion.

$$V_{ij} = \sum_{q \in S_q} \vec{K}_i(q) \cdot \vec{K}_j(q) \tag{9}$$

The sign of $V_{ij}$ indicates the relative polarity between the $i^{\text{th}}$ and $j^{\text{th}}$ stator poles. If $V_{ij}$ is positive, there is a tendency for the $i$–$j$ pair to produce torque vectors in similar directions; thus, they should be connected with the same polarity. A negative value for $V_{ij}$ indicates the tendency for the pair to produce antagonistic torque vectors and indicates that the pair should should be assigned opposing polarities. Applying this technique to the phase groupings in Figure 12 results in the polarities shown in Figure 13. This figure shows the inner surface of the rotor bowl, with individual stator poles labeled with a color corresponding to their phase grouping and north or south corresponding to their polarity. Here, "N" indicates that the north pole of the electromagnet points inwards, according to the direction of the conventional current flow and the right-hand rule.

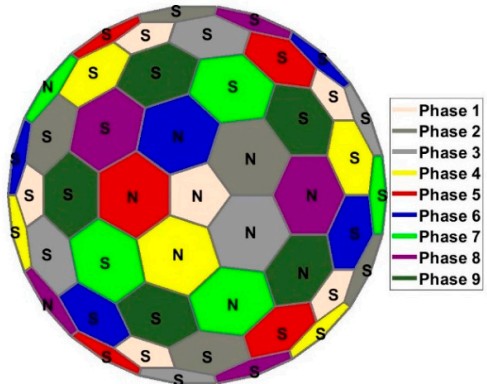

**Figure 13.** Stator phases and polarities. The phase groupings of Figure 12, together with Equation (9), produce these groupings and polarities. "N" and "S" correspond to north and south polarities, respectively, on the inner surface of the rotor bowl, as determined by the conventional current right-hand rule.

### 3. Results

The initial PMSSM controller development involved manually tuning the individual phase current controllers. The PID loops for the motor phases were tuned using Ziegler Nichols' heuristic approach [34]. The current controllers exhibited reasonable responses with a rise time of 20 ms, although tracking errors were observed below 50 mA, due to limited current sensing at lower magnitudes.

Subsequently, the step response of the PMSSM was evaluated using step commands of 20°, 30° and 90° for the $\alpha$, $\beta$ and $\gamma$ angles, respectively. Figure 14 shows the response of the PMSSM to the step input. The steady state errors were 0.25°, 0.44° and 0.16° for the $\alpha$, $\beta$ and $\gamma$ angles, respectively. The data were collected with MATLAB (Mathworks Inc., Natick, MA, USA) from the MCU using serial communication with a sampling rate of 50 Hz.

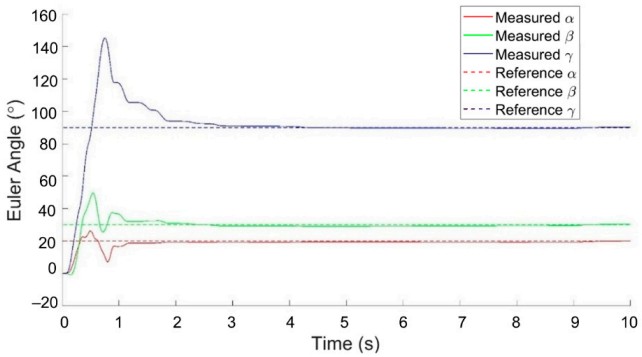

**Figure 14.** Experimental closed-loop results: typical PID responses for step references in $\alpha$, $\beta$ and $\gamma$ angles.

Separate testing was conducted to track time-varying references about multiple axes. The reference trajectory for $\alpha$ was sinusoidal, with an amplitude of $30°$ and a frequency of $72°/s$, while the reference for $\gamma$ was a ramp with a slope of $72°/s$. The setpoint for $\beta$ remained fixed at 0. Figure 15a provides snapshots of the first 5.0 s of this rotor reference trajectory. Figure 15b shows the tracking performance for the time-varying reference. The average absolute tracking error was $3.10°$, $3.94°$ and $6.38°$ for $\alpha$, $\beta$ and $\gamma$, respectively, while the maximum tracking error was $10.56°$, $19.70°$ and $14.30°$. The associated phase currents are shown in Figure 15c. Phase currents were saturated at 1.5 A to prevent damage to the phase windings.

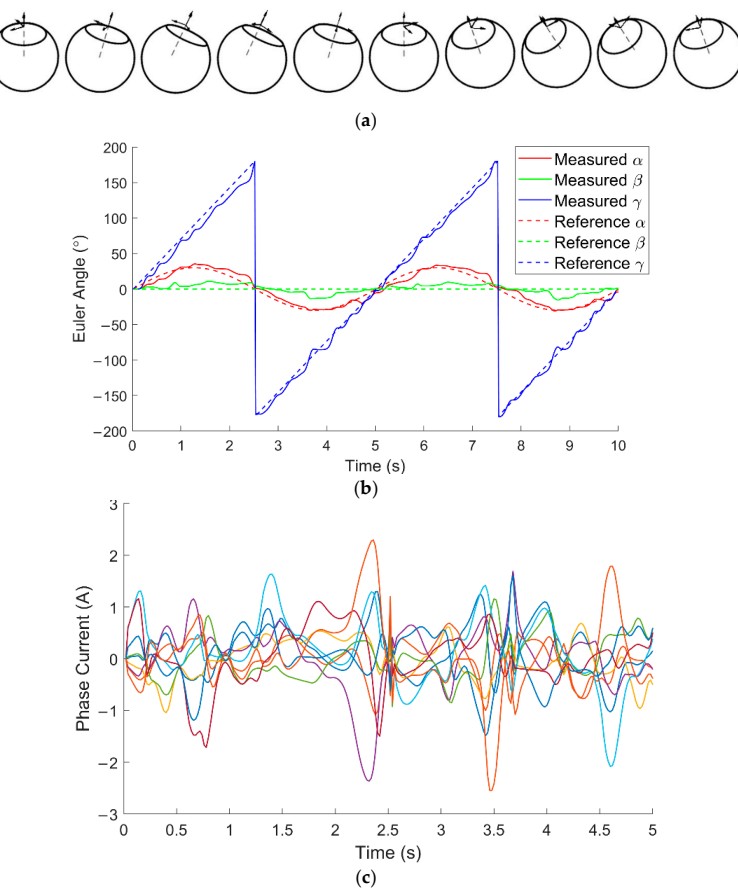

**Figure 15.** Experimental closed-loop results: (**a**) Geometric visualization of the rotor reference trajectory; (**b**) Time responses for reference tracking in $\alpha$, $\beta$ and $\gamma$ angles; (**c**) Corresponding phase currents. While the average absolute tracking error is low for b, periodic spikes in the error are present, predominantly with respect to the $\gamma$ angle.

## 4. Discussion and Conclusions

While the PMSSM successfully tracked both step and time-varying reference trajectories, it did so with a relatively large overshoot (up to 55.13° for the step responses of Figure 14), long settling times (up to 2.98 s), and large tracking errors (a mean of 4.47° and a peak of 19.70° for the time-varying reference responses of Figure 15). Numerous factors contributed to this suboptimal performance, most notably limitations in the control algorithm (which did not account for rotor dynamics or stator back-iron) and stick-slip behavior, resulting from the friction between the surface of the rotor and the Hudson bearings. These factors caused the PMSSM orientation to overshoot and oscillate about the setpoint, as evident in Figure 15a. Inaccuracies in the electromechanical model, which did not account for the stator back-iron when computing the characteristic torque vector, resulted in discrepancies between the actual and commanded torque vector. Additionally, torque generated by the PM/stator-pole interaction near the edges of the bowl differed significantly from the torque generated away from the edges, due to differences in the reluctance of the magnetic circuits. Eddy current losses also contributed to performance limitations by producing additional torque, which opposes the rotor's motion. The sampling time of the main control loop was constrained to 50 Hz by the computationally expensive calculation of the characteristic torque matrix, further restricting the bandwidth of the motor. A higher sampling rate (which could be achieved using a more sophisticated microcontroller) would help dampen high-frequency signal oscillations about the desired orientations, as observed in Figure 15a. The IMU was susceptible to drift over time (1.5°/h), inducing errors in calculating the orientation of the rotor as experiments were repeated over time.

Maintaining a uniform air gap between the stator and the rotor using multiple Hudson bearings was challenging. For certain rotor orientations, some bearings were not in contact with the rotor, while other bearings were overloaded.

The asymmetric rotor, overloaded bearings, and inadequate hardness of the epoxy rotor caused dents on the rotor surface, created "stick-slip" friction, and distorted the torque vector, making the motor more difficult to control. Machining the epoxy layer to produce a more spherical and smooth rotor was also problematic. Air bubbles formed while casting the epoxy layer persisted even after degassing the mold during the casting process and appeared as dents on the surface of the rotor after CNC turning. To address these defects, the surface was coated with another thin layer of epoxy and hand sanded. Maintaining a spherical shape and smooth surface throughout this process was challenging, as even small deviations in roundness caused variations in rotation. Future refinements to the PMSSM design should focus on improved bearings and enhanced rotor materials (with appropriate mechanical and magnetic properties) and fabrication methods (that ensure spherical geometry and uniform bearing support).

This research addresses many of the design and fabrication challenges associated with spherical motors, including selecting appropriate methods for orientation sensing and optimizing the distribution of PMs within the rotor and stator phases. However, the optimization algorithms used to govern this design process could be improved in future research. Developing a more sophisticated system model, one that accounts for rotordynamics, bearing friction and actuator nonlinearities, and using this model for nonlinear control synthesis would address the issues of stick-slip and significantly improve tracking performance. Ongoing research is investigating the development of such models, and their utilization in the development of more sophisticated control strategies (including multivariable, nonlinear, and intelligent controllers).

**Author Contributions:** Conceptualization, J.A.S. and G.D.B.; methodology, J.A.S., S.R.M. and G.D.B.; software, J.A.S.; validation, J.A.S., S.R.M. and S.R.J.; formal analysis, J.A.S.; investigation, J.A.S. and S.R.M.; resources, G.D.B. and S.R.J.; data curation, J.A.S.; writing—original draft preparation, J.A.S. and S.R.M.; writing—review and editing, G.D.B. and S.R.J.; visualization, J.A.S.; supervision, G.D.B.; project administration, G.D.B. All authors have read and agreed to the published version of the manuscript.

**Funding:** This research received no external funding.

**Institutional Review Board Statement:** Not applicable.

**Informed Consent Statement:** Not applicable.

**Data Availability Statement:** Data supporting the reported results may be obtained via the corresponding author.

**Acknowledgments:** Machining of PMSSM prototypes was completed by Steve Cameron and Vincent Chicarelli of the Mechanical & Aerospace Engineering Research Fabrication Facility.

**Conflicts of Interest:** The authors declare no conflict of interest.

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
