# Peer review of "A Permanent Magnet Synchronous Spherical Motor for High-Mobility Servo-Actuation"

_machines, doi:10.3390/machines10080612_

Round 1
Reviewer 1 Report
Dear Authors,
your article is interesting, but I would like to see more waveforms derived from simulation and experiment, for comparison purposes.
What are the advantages of this type of machine compared to conventional machine? Please analyze it more in the introduction section.
Author Response
The authors of Machines-1809647 (“A Permanent Magnet Synchronous Spherical Motor for High-Mobility Servo-Actuation”) would like to thank Reviewer 1 for the thorough evaluation of our manuscript and the constructive feedback. We have carefully considered each reviewer comment, and have addressed each as summarized below:
Comment: Your article is interesting, but I would like to see more waveforms derived from simulation and experiment, for comparison purposes.
Response: The authors agree that providing simulated and experimental waveforms for comparison purposes would be an interesting addition to our paper. While we have modeled and simulated the dynamics of this motor, the simulated and experimental waveforms did not compare favorably due to unmodelled dynamics (arising from “stick-slip” friction, rotor eccentricity, whole-stator magnetostatics, etc.) which are very difficult to model accurately.
Comment: What are the advantages of this type of machine compared to conventional machine? Please analyze it more in the introduction section.
Response: This is an excellent recommendation. In our revised manuscript, we have better detailed the advantages of this spherical motor and provided several examples of its application (with newly added citations). A revised paragraph in the Introduction now reads: “As servomechanism technologies and applications continue to advance, there is a growing need for innovative actuators with increased kinematic mobility and power density. Servomechanisms with multiple degrees-of-freedom (DOF) are typically realized using single-DOF joints (rotational and/or prismatic), arranged in series or parallel and powered by multiple single-DOF actuators. These configurations can be accompanied by kinematic singularities, backlash, limited power density and efficiency, and computationally expensive inverse kinematics. The development of direct-drive spherical motors offers potential means to overcome these limitations by replacing multiple single-DOF actuators with a single multi-DOF actuator. Potential applications include satellite attitude control (Kim, et al., 2014), encoder, camera and scanner gimbals (Kaneko, Yamada and Itao, 1989), rotational stages for CNC machining and additive manufacturing (Bai, et al., 2018), haptic feedback devices (Bai, et al., 2012), and robotic actuation (Ciupitu, et al., 2003; Kumagai and Hollis, 2013).”
Reviewer 2 Report
The presented structure is very interesting, mechanical realization of the assembly is a great challenge, it is well explained, but, finally, we ask ourselves the question of the interest of the approach, is it not only a style exercise?
Authors would benefit from presenting one or two applications which would require such a complex mechanical structure and, above all, explaining how a spherical structure is more advantageous than the mechanical association of two rotating machines with crossed axes, for limited strokes.
I do not understand the dimensioning carried out in paragraph 2.1, the calculated magnetic field is an armature reaction field which does not participate of torque production, the proximity of the interior plates will create a short circuit for the flux of the permanent magnets, it would have been necessary, at least, to bevel them. Moreover, the magnets being quite far from each other, it would have been necessary to position a magnetic yoke under them, the absence of this yoke means that the field produced in the air gap will be relatively weak. However, it will be difficult to modify the prototype in this way. The electromagnetic dimensioning of the device does not seem very judicious, the motor torque produced is necessarily very low, which explains a large part of the setbacks encountered during the experiment.
There is extremely little chance that the epoxy resin, thrust ball bearing interface will be able to operate over time, grooves will form which will prevent any operation. But deposition of a hard coating on the rotor is very difficult to achieve.
Author Response
The authors of Machines-1809647 (“A Permanent Magnet Synchronous Spherical Motor for High-Mobility Servo-Actuation”) would like to thank Reviewer 2 for the thorough evaluation of our manuscript and the constructive feedback. We have carefully considered each reviewer comment, and have addressed each as summarized below:
Comment: The presented structure is very interesting, mechanical realization of the assembly is a great challenge, it is well explained, but, finally, we ask ourselves the question of the interest of the approach, is it not only a style exercise? Authors would benefit from presenting one or two applications which would require such a complex mechanical structure and, above all, explaining how a spherical structure is more advantageous than the mechanical association of two rotating machines with crossed axes, for limited strokes.
Reply: This is an excellent point. In our revised manuscript, we have better detailed the advantages of this spherical motor and provided several examples of its application (with newly added citations). A revised paragraph in the Introduction now reads: “As servomechanism technologies and applications continue to advance, there is a growing need for innovative actuators with increased kinematic mobility and power density. Servomechanisms with multiple degrees-of-freedom (DOF) are typically realized using single-DOF joints (rotational and/or prismatic), arranged in series or parallel and powered by multiple single-DOF actuators. These configurations can be accompanied by kinematic singularities, backlash, limited power density and efficiency, and computationally expensive inverse kinematics. The development of direct-drive spherical motors offers potential means to overcome these limitations by replacing multiple single-DOF actuators with a single multi-DOF actuator. Potential applications include satellite attitude control (Kim, et al., 2014), encoder, camera and scanner gimbals (Kaneko, Yamada and Itao, 1989), rotational stages for CNC machining and additive manufacturing (Bai, et al., 2018), haptic feedback devices (Bai, et al., 2012), and robotic actuation (Ciupitu, et al., 2003; Kumagai and Hollis, 2013).”.
Comment: I do not understand the dimensioning carried out in paragraph 2.1, the calculated magnetic field is an armature reaction field which does not participate of torque production, the proximity of the interior plates will create a short circuit for the flux of the permanent magnets, it would have been necessary, at least, to bevel them. Moreover, the magnets being quite far from each other, it would have been necessary to position a magnetic yoke under them, the absence of this yoke means that the field produced in the air gap will be relatively weak. However, it will be difficult to modify the prototype in this way. The electromagnetic dimensioning of the device does not seem very judicious, the motor torque produced is necessarily very low, which explains a large part of the setbacks encountered during the experiment.
Reply: The authors agree with the Reviewer, and more could be done in future design iterations to address these comments. In the revised manuscript, we have addressed the comments related to the dimensioning, and that our FEA-simulated stator fields do not participate in torque production. In our revised manuscript, we have better explained the utility of these simulations in dimensional design optimization: “To optimize these stator pole dimensions, three-dimensional magnetostatic simulations were conducted using finite element analysis (FEA) software (ANSYS Electronics Desktop, Canonsburg, PA, USA). Each simulation utilized more than 300,000 tetrahedral elements, with curvilinear meshing automatically generated. Parametric sweeps quantified the nonlinear relationships between magnetic flux density and stator pole dimensions, most notably stem diameter d and length l. Figure 3 shows FEA results for adjacent stator poles simultaneously excited with the maximum 300 A-t; case 3b (d= 12 mm) effectively addresses the tradeoff between stator stem mass and magnetic saturation. While the stator pole excitations used in these simulations do not necessarily correspond to those used for rotor torque production, they provided useful insights into dimensional optimization.”
Comment: There is extremely little chance that the epoxy resin, thrust ball bearing interface will be able to operate over time, grooves will form which will prevent any operation. But deposition of a hard coating on the rotor is very difficult to achieve.
Reply: We agree with these comments, and have addressed them in greater detail in the revised manuscript: “Maintaining a uniform air gap between the stator and the rotor using multiple Hudson bearings was challenging. For certain rotor orientations, some bearings were not in contact with the rotor, while other bearings were overloaded. The asymmetric rotor, overloaded bearings, and inadequate hardness of the epoxy rotor caused dents on the rotor surface, created “stick-slip” friction, and distorted the torque vector, making the motor more difficult to control. Machining the epoxy layer to produce a more spherical and smooth rotor was also problematic. Air bubbles formed while casting the epoxy layer persisted even after degassing the mold during the casting process and appeared as dents on the surface of the rotor after CNC turning. To address these defects, the surface was coated with another thin layer of epoxy and hand sanded. Maintaining a spherical shape and smooth surface throughout this process was challenging, as even small deviations in roundness caused variations in rotation. Future refinements to the PMSSM design should focus on improved bearings and enhanced rotor materials (with appropriate mechanical and magnetic proper-ties) and fabrication methods (that ensure spherical geometry and uniform bearing support).”.
Reviewer 3 Report
The article is very interesting and useful according to the current world trend in PM machines. The spherical machine model is described in detail. The Authors have correctly used ANSYS software to run the magnetostatic simulation. The article is written in an appropriate way and clearly presents the obtained results. The Authors use custom-made controller board includes a microcontroller, servo drivers, a wireless serial interface, and a USB PC interface, what will be used in further researches to optimize applied algorithms, as mentioned in conclusion part. In my opinion this topics can be further develop according to PM spherical motors, I am very impressed with this work. The conclusion part contains necessary summary.
I would suggest the Authors to address just this MINOR point, interesting from the field analysis point of view:
- The Authors are invited to mention about the mesh quality in ANSYS model (how many elements? Default or users mesh?).
Author Response
The authors of Machines-1809647 (“A Permanent Magnet Synchronous Spherical Motor for High-Mobility Servo-Actuation”) would like to thank Reviewer 3 for the thorough evaluation of our manuscript and the constructive feedback. We have carefully considered each reviewer comment, and have addressed each as summarized below:
Comment: The article is very interesting and useful according to the current world trend in PM machines. The spherical machine model is described in detail. The Authors have correctly used ANSYS software to run the magnetostatic simulation. The article is written in an appropriate way and clearly presents the obtained results. The Authors use custom-made controller board includes a microcontroller, servo drivers, a wireless serial interface, and a USB PC interface, what will be used in further researches to optimize applied algorithms, as mentioned in conclusion part. In my opinion this topics can be further develop according to PM spherical motors, I am very impressed with this work. The conclusion part contains necessary summary.
I would suggest the Authors to address just this MINOR point, interesting from the field analysis point of view: the Authors are invited to mention about the mesh quality in ANSYS model (how many elements? Default or users mesh?).
Reply: We appreciate the Reviewer’s supportive comments, and agree that more detail regarding the FEA simulations would be helpful. In response to this recommendation, we have added the following sentences to our revised manuscript: ” To optimize these stator pole dimensions, three-dimensional magnetostatic simulations were conducted using finite element analysis (FEA) software (ANSYS Electronics Desktop, Canonsburg, PA, USA). Each simulation utilized more than 300,000 tetrahedral elements, with curvilinear meshing automatically generated. Parametric sweeps quantified the nonlinear relationships between magnetic flux density and stator pole dimensions, most notably stem diameter d and length l.”